# Construction of Artificial Ovaries with Decellularized Porcine Scaffold and Its Elicited Immune Response after Xenotransplantation in Mice

**DOI:** 10.3390/jfb13040165

**Published:** 2022-09-28

**Authors:** Tong Wu, Yue-Yue Gao, Xia-Nan Tang, Jin-Jin Zhang, Shi-Xuan Wang

**Affiliations:** 1National Clinical Research Center for Obstetrical and Gynecological Diseases, Tongji Hospital, Tongji Medical College, Huazhong University of Science and Technology, Wuhan 430030, China; 2Key Laboratory of Cancer Invasion and Metastasis, Ministry of Education, Tongji Hospital, Tongji Medical College, Huazhong University of Science and Technology, Wuhan 430030, China; 3Department of Obstetrics and Gynecology, Tongji Hospital, Tongji Medical College, Huazhong University of Science and Technology, Wuhan 430030, China

**Keywords:** artificial ovaries, decellularization, ovarian function, immune response

## Abstract

Substitution by artificial ovary is a promising approach to restore ovarian function, and a decellularized extracellular matrix can be used as a supporting scaffold. However, biomimetic ovary fabrication and immunogenicity requires more investigation. In this study, we proposed an effective decellularization protocol to prepare ovarian scaffolds, which were characterized by few nuclear substances and which retained the extracellular matrix proteins. The ovarian tissue shape and 3-dimensional structure were well-preserved after decellularization. Electron micrographs demonstrated that the extracellular matrix fibers in the decellularized group had similar porosity and structure to those of native ovaries. Semi-quantification analysis confirmed that the amount of extracellular matrix proteins was reduced, but the collagen fiber length, width, and straightness did not change significantly. Granulosa cells were attached and penetrated into the decellularized scaffold and exhibited high proliferative activity with no visible apoptotic cells on day 15. Follicle growth was compromised on day 7. The implanted artificial ovaries did not restore endocrine function in ovariectomized mice. The grafts were infiltrated with immune cells within 3 days, which damaged the artificial ovary morphology. The findings suggest that immune rejection plays an important role when using artificial ovaries.

## 1. Introduction

The ovary is a critical organ for steroid hormone production and female gamete generation. However, it may be disturbed by aging, tumors, the environment, and medical interventions, which eventually lead to infertility and menopausal complications [1]. Although hormone replacement therapy has been implemented clinically for decades to compensate for a lack of sex hormones, it may cause adverse effects, such as venous thromboembolism, stroke, and hormone-sensitive cancers [2]. As many more people choose to postpone pregnancy, fertility preservation has also become an increasing concern. Artificial ovaries were designed to achieve hormone secretion and egg release simultaneously to serve as natural ovaries. However, the artificial ovary field remains in the fundamental stages of exploration and more evidence is needed.

Artificial ovary structure and morphology vary based on the bioengineering approach, which could be as simple as a drop of alginate hydrogel or manipulation via bioprinting [3]. Overall, artificial ovaries are mainly composed of supporting scaffolds and ovarian cells, which include follicles [4]. In the former, synthetic polymers can be customized to suitable degradation and mechanical properties. However, they may cause inflammation or a rejection response in the host. By contrast, natural polymers include collagen, fibrin, and plasma clots and have the advantages of biocompatibility and cytotoxicity. Nevertheless, neither synthetic nor natural materials can thoroughly mimic the spatial distribution, vascularization, or mechanical property of ovaries. It is challenging to reproduce the native ovarian extracellular matrix (ECM).

The advancement of ECM biology and new biotechnologies have enabled the introduction of decellularized (Dec) ECM to construct artificial organs [5]. In Dec tissues, immunogenic substances are removed while ECM proteins are retained, as are their ultrastructural architecture and mechanical properties. In this way, Dec scaffolds can not only provide cell attachment support but also promote cell growth. Decellularized porcine corneal grafts have been used to treat fungal corneal infection in a clinical trial, with high cure rate and no recurrence of infection [6]. Small intestinal submucosa of pigs has been made into various commercially available products to repair blood vessels, skin, the diaphragm, pericardium and even infarcted myocardium [7]. In a human relevant clinical model, human-scale whole lungs were created using porcine acellular lung scaffolds and could survive as long as 2 months after porcine transplantation [8]. However, the application of Dec tissues in the field of the female reproduction system is preliminary.

Laronda et al. first fabricated ovarian-specific Dec tissues in 2015 [9]. They successfully decellularized human and bovine ovaries and constructed artificial ovaries, which produced estradiol and supported follicular development in vivo [9]. Since then, ovarian decellularization has been performed in other species, such as mouse, pig, and sheep [10,11]. Moreover, multiple cell types, which include human pre-antral follicles, female germline stem cells, mesenchymal stem cells, and dermal fibroblasts, have been loaded on Dec tissues [12,13,14]. In sum, the use of Dec scaffolds, with or without seeding cells, was reported to differently promote follicular viability and increase hormonal secretion [15]. However, most of these studies were limited to in vitro evaluation. More in-depth studies are needed to confirm the efficiency of artificial ovaries in vivo.

To investigate Dec ovarian scaffold fabrication and immunogenicity, we decellularized porcine ovaries and comprehensively evaluated their feasibility and efficiency. We detected the residual DNA content and ECM proteins to validate our method. Murine ovarian granulosa cells (GCs) and follicles were seeded onto the scaffolds to construct preliminary artificial ovaries in vitro. Then, the function and histology were examined. The results suggested that Dec ovarian scaffolds could be appropriate materials for constructing artificial ovaries, but attention should be paid to alleviate the immune response.

## 2. Materials and Methods

### 2.1. Pretreatment of Porcine Ovarian Tissues

Porcine ovaries (*n* = 30) were obtained from a local slaughterhouse and transported to the laboratory within 3 h in phosphate-buffered saline (PBS) solution. Upon arrival, the surplus blood, muscle, fascia, and adipose tissues were removed. Some ovaries (*n* = 5) were randomly cryopreserved or fixed in 4% (*w/v*) paraformaldehyde to be used as the control. The remaining ovaries (*n* = 25) were frozen at −80 °C for subsequent decellularization. Follicular fluid was released by pricking with a 1 mL insulin syringe, then the ovaries were opened along the hilus and 2 mm thick cortical tissues were carefully separated from the medulla. The cortex was cut into 5 mm × 5 mm × 2 mm pieces and underwent an additional round of pricking with the syringe.

### 2.2. Decellularization and Scaffold Preparation

The tissues were decellularized using a protocol that combined mechanical, chemical, and biological methods. In the mechanical portion, the tissues were repeatedly frozen at −80 °C and thawed at room temperature three times. Then, they were shaken in double distilled water (ddH_2_O) for 6 h with an exchange of at least three times to remove blood and follicular fluid. In the chemical portion, the tissues were immersed and shaken in a combined solution of 2% (*w/v*) sodium deoxycholate (SDC, D8331, Solarbio, Beijing, China) and 4% (*v/v*) Triton X-100 (0694, Amresco, Solon, OH, USA) for 36 h with an exchange every 6 h. Then, they were shaken for an additional 36 h in 1% (*w/v*) sodium dodecyl sulfate (SDS, BioFroxx, Einhausen, Germany). In the biological portion, the tissues were incubated in RNase/DNase solution (80 U/mL) (Servicebio, Wuhan, China) for 6 h at 37 °C. Finally, the tissues were washed in ddH_2_O to eliminate residual detergent and cellular debris every 6 h for up to 24 h. The whole ovaries were decellularized by treatment with 2% SDC/4% Triton X-100 and 1% SDS for at least 2 weeks. The tissues were sterilized with 4-h treatment with 0.1% peracetic acid/4% ethanol (Kemiou, Tianjin, China).

### 2.3. Comparisons of Different Ovarian Decellularization Strategy

To compare the efficiency of different decellularization methods, we conducted a systematic search in electronic medical databases: PubMed, Embase, Web of Science, and The Cochrane Central Register of Controlled Trials until 8 September 2022. The search keywords were: “decellularization” or “acellular” or “recellularization” AND “ovary” or “ovarian tissue” or “follicle”. In addition, we reproduced some previously reported decellularization protocols [9,13,16]. These protocols were applied to human, bovine and sheep species. The strategies of Laronda et al., Mirzaeian et al., and Pors et al. were indicated as “M”, “L” and “S”, respectively. In brief, “M” includes shaking 0.5-mm-thick ovarian cortex in 0.1% SDS for 24 h [13]. “L” includes shaking 2-mm-thick ovarian cortex in 0.5 M NaOH overnight [9]. “S” includes treating samples with 0.1% SDS for at least 3 h [16].

### 2.4. Scanning Electron Microscopy (SEM)

The decellularized samples were lyophilized overnight, fixed in 2.5% glutaraldehyde for 2 h, and kept in a 4 °C refrigerator. The next day, the samples were rinsed with PBS overnight, dehydrated in 35–55–75–85–95% graded ethanol–water for 15 min per rinse and treated with isoamyl acetate for 15 min. The dehydrated samples were mounted on conductive carbon tape and sprayed with gold particles. Then, the morphology and structure were acquired using an ultra-high resolution scanning electron microscope (HITACHI Regulus 8100, Tokyo, Japan).

### 2.5. Connective Tissue Staining and Morphological Evaluation

Native and decellularized samples were fixed in 4% paraformaldehyde, dehydrated in graded alcohol series, embedded in paraffin, and sectioned into 5-μm sections. The glass slides were stained with hematoxylin and eosin (H&E, Servicebio, Wuhan, China), Masson trichrome, periodic acid–Schiff, Alcian blue, and picrosirius red (PSR) according to the manufacturer’s instructions. All reagents were purchased from Servicebio. The sections were imaged on a light microscope (CX43, Olympus, Tokyo, Japan). PSR-stained slides were assessed by phase-contrast microscopy (Eclipse Ci, Nikon, Tokyo, Japan) and analyzed by the CT-FIRE program.

### 2.6. Immunohistochemistry Staining and Semi-Quantitative Analysis

Both structural and soluble ECM proteins were evaluated to observe the integrity of the Dec scaffolds. Antibodies against COL1A1 (BA0325, Boster, Pleasanton, CA, USA), COL2A1 (BA0533, Boster), COL3A1 (PB0125, Boster), COL4A1 (A10710, ABclonal, Wuhan, China), FN (GB112093, Servicebio, Wuhan, China), laminin (NB300-144, Novus, Littleton, CO, USA), AMH (14461-1-AP, Proteintech, Chicago, IL, USA), TGF-β (GB111876, Servicebio), BMP15 (A01842, Boster), CTGF (BA0752, Boster), TOMM2 (sc-17764, Santa cruz, Dallas, TX, USA), COX2 (A1253, ABclonal, Wuhan, China), RPS6 (AP0637, ABclonal, Wuhan, China) and α-TUBULIN (66031-1-Ig, Proteintech, Chicago, IL, USA) were diluted to the required concentrations and visualized with ready-to-use SABC-AP (streptavidin–biotin–alkaline phosphatase) kits (SA1052, Boster, Pleasanton, CA, USA) and a DAB (diaminobenzidine) chromogenic kit (AR1027, Boster, Pleasanton, CA, USA). Antigen retrieval was performed by two consecutive incubations in citric acid solution in boiling water for 5 min each, followed by cooling at room temperature and rinsing with PBS three times. Blocking against nonspecific binding was done using 10% goat serum solution for 30 min at 37 °C. Slides were incubated with the primary antibodies overnight. The next day, the slides were incubated with secondary antibodies and SABC solution for 1 h and 30 min at 37 °C, respectively. Finally, the slides were stained with DAB and placed in hematoxylin for 3 min. Immunohistochemistry was performed on 3–5 sections per sample and at least three samples per group. Each experiment was performed ≥ 2 times and included no-primary antibody controls. The semi-quantitative results were analyzed using ImageJ (V1.53a, NIH) software.

### 2.7. Collagen Content Detection

The collagen contents were quantified using a hydroxyproline detection kit (A030-2, Jiancheng, China) according to the manufacturer’s instructions. Hydroxyproline is specifically detected in collagens and can reflect the collagen content. Briefly, a 30 mg (wet weight) sample was homogenized in hydrolysate solution and solubilized for 20 min at 100 °C. Then, we adjusted the pH to 6.5, added anticarbon to the samples, and centrifuged them. The supernatant was retained and the absorbance was recorded at 555 nm wavelength. The experiments were performed in triplicate.

### 2.8. DNA Extraction, Quantitation, and Electrophoresis

For the DNA quantification using a DNA extraction kit (DP304, TIANGEN, Beijing, China), the samples were completely homogenized and solubilized in 200 μL buffer and digested overnight with proteinase K in a 56 °C water bath. The DNA was precipitated from the hydrous phase with 100% ethanol, after which the extracts were washed with 70% ethanol. After dissolving the resulting pellet in RNase-free water, the DNA concentration was determined using a spectrophotometer. The amount of DNA was averaged from a set of three independent runs and expressed as the μg/mg dry weight of the samples.

The size of the remaining DNA fragments in the Dec and control ovaries were assessed with electrophoresis. A 2% (*w/v*) agarose gel was run in Tris base–acetic acid–ethylenediaminetetraacetic acid (TAE) followed by the addition of 2 μL 4S Red Plus (A606695, Sangon, Wuhan, China). Each extracted DNA sample (5 µL) was loaded into each well. The reference was a 50-bp DNA ladder (TSJ050-500, TSINGKE, Beijing, China). After running the gel (110 V, 30 min), the DNA was visualized and photographed using a ChemiDoc™ Imaging System (Bio-Rad, Hercules, CA, USA).

### 2.9. Isolation, Culture, and Identification of GCs and Follicles

All mice in this study were obtained from the Hubei Center for Disease Control and Prevention Animal Center and were treated in accordance with the university ethics committee guidelines on standard animal care. The protocol was approved by the Ethics Committee of Tongji Hospital (Approval ID: TJ-IRB20210319). For GCs isolation, 4-week-old female C57BL/6J mice were euthanized by cervical dislocation and their ovaries were removed by dissection in preincubated Leibovitz’s 15 medium supplemented with 100 μg/mL streptomycin and 100 IU/mL penicillin. Ovaries were punctured with a needle to release the GCs. The GC suspensions were then centrifuged at 800 rpm for 5 min and re-suspended in corresponding culture medium. For follicles isolation, 14- to 21-day-old female C57BL/6J mice were euthanized by cervical dislocation. To obtain preantral follicles for in vitro experiment, follicles with a diameter of 100–180 μm, an intact basal membrane, a central and spherical oocyte, and surrounding GCs were selected. To obtain follicles of various stages (primordial, primary and preantral follicles) for in vivo experiment, follicles with a diameter of 20–180 μm and a central and spherical oocyte were selected for seeding. The diameters of primordial, primary and preantral follicles are <30 μm, 30–80 μm and 80–250 μm, respectively. 

The culture medium for GCs was McCoy’s 5A supplemented with 10% fetal bovine serum (FBS), 10 ng/mL androgen, 100 μg/mL streptomycin and 100 IU/mL penicillin. The growth medium for follicles was α-MEM supplemented with 10% FBS, 10 mIU/mL recombinant follicle stimulating hormone (8576-FS, R&D systems, Minneapolis, MN, USA), 100 μg/mL streptomycin, 100 IU/mL penicillin and ITS (insulin 10 μg/mL, transferrin 5.5 μg/mL, selenium 6.7 ng/mL, Sigma-Aldrich, St. Louis, MO, USA). The cells were incubated at 37 °C with 5% CO_2_.

### 2.10. Construction of Artificial Ovaries

Each scaffold was recellularized with the GCs and follicles by repeated pipetting on the surfaces in 96-well culture plates to construct artificial ovaries using the Dec scaffolds (Appendix A). The optimal density of the ovarian somatic cells was 2.0 × 10^6^. The sum of follicles was appropriately 80 per scaffold. Each artificial ovary (*n* = 10 per group) was placed for 30 min in the incubator to allow cell attachment. Subsequently, additional cell culture medium was added. The artificial ovaries were incubated in Dulbecco’s modified Eagle’s F12 medium containing 10% fetal bovine serum, 1% penicillin/streptomycin, 1% non-essential amino acid (Gibco, Grand Island, NY, USA), 1% L-glutamine (Gibco, Grand Island, NY, USA), and 1% insulin-transferrin-selenium (ITS; Sigma-Aldrich, St. Louis, MO, USA). For the in vivo experiments, the constructs were incubated for 1 day and underwent subcutaneous transplantation the next day.

### 2.11. Terminal Deoxynucleotidyl Transferase-Mediated dUTP Nick end-Labeling (TUNEL) Assay

Apoptotic cells were evaluated with a TUNEL detection kit according to the manufacturer’s instructions (C1089, Beyotime, Shanghai, China). Dewaxed and rehydrated sections were digested with 10 µg/mL proteinase K for 30 min at 37 °C. Prior to the labelling procedure, positive controls were treated with recombinant DNase I for 10 min to induce DNA strand breaks. All samples were counterstained with DAPI (4′,6-diamidino-2-phenylindole) and the slides were observed under an Olympus BX53 fluorescence microscope.

### 2.12. Animal Model

Eight-week-old female C57BL/6J mice (*n* = 40) were used as the host and divided randomly into sham-operated (Sham), ovariectomized (OVX), scaffold-only (blank), and artificial ovary (Dec scaffolds containing follicles [graft]) groups (*n* = 10 each group) (Appendix A). All animals were anesthetized with ketamine/xylazine (80/5 mg/kg body weight) and bilateral ovariectomy was performed in all but the Sham group. The vaginal orifices of the mice were checked daily. Finally, the grafts were removed and fixed in 4% paraformaldehyde for analyses.

### 2.13. In Vivo Assessment of Artificial Ovaries

For transplantation, the lateral abdominal wall was shaved, disinfected, and opened with a pair of scissors. The artificial ovaries were placed between the skin and sub-serosal fascia attached firmly beneath the peritoneum. Then, the opened layers and abdominal wall were closed with absorbable sutures (6-0). The mice were kept in individual cages with adequate food and water under standard conditions.

### 2.14. Estrous Cycle Examination

Vaginal smears were performed under light microscopy. The phase of estrous cycle was determined based on the proportions of nucleated and keratinized epithelial cells and leukocytes. Estrous cycle disorder is a distinguishing characteristic of ovarian function failure.

### 2.15. Statistical Analysis

The data were statistically analyzed using GraphPad Prism v8 (GraphPad Software, San Diego, CA, USA). The different groups were compared using analysis of variance (ANOVA) followed by the Tukey test (as a post hoc test). All data were precisely defined as the mean and standard error of the mean (mean ± standard error of mean). In all analyses, *p* ≤ 0.05 was considered statistically significant.

## 3. Results

### 3.1. Decellularization of Porcine Ovaries to Acquire Dec Scaffolds

Porcine ovarian samples were chosen for their easy accessibility and large volume. We eliminated the cellular components in the porcine ovaries to obtain Dec ECM scaffolds. We separated the ovarian medulla and cortex, then sliced them into 5 mm × 5 mm × 2 mm strips (Figure 1A). The strips were sequentially decellularized and their color changed from pink to white while the shape and 3-dimensional structure were well-preserved. H&E staining demonstrated the successful removal of the cellular remnants and nuclei from the Dec tissues, while cells were clearly observed in the native tissues (Figure 1B). Similarly, no obvious fluorescence was observed in the DAPI-stained Dec tissues when compared with native tissue (Figure 1C). The DNA assay confirmed a significantly lower amount of double-stranded DNA (12.86 ± 1.707 ng/mg, *p* < 0.0001) after decellularization (Figure 1D) and the DNA fragments were <200 bp (Figure 1E). We also compared the efficiency of different decellularization protocols through a literature search (Appendix A). Therefore, the Dec scaffolds met the previously established decellularization criteria. We applied the protocol to entire ovaries to test the efficiency of our decellularization approach (Figure 1F). H&E staining revealed the absence of basophilic materials from both the cortical and medulla regions (Figure 1G). Ultimately, we successfully obtained Dec scaffolds using ovarian cortex and whole tissue.

### 3.2. Evaluation of Dec Scaffolds via Residual ECM Protein Component Detection

We performed SEM, connective tissue staining, and immunohistochemistry to evaluate the type, quantity, and ultrastructural architecture of ECM proteins in the Dec ovaries. Some ECM proteins significantly reduced in Dec tissues (Appendix A). SEM assessment indicated the complex fiber network integrity in the Dec scaffolds (Figure 2A). Consistent with the light microscopy findings, the cavity microstructures were previously occupied by parenchymal cells and were devoid of cells after the decellularization. Masson trichrome staining revealed the absence of red-stained myocytes and red blood cells after decellularization (Figure 2A). Collagen fibers were maintained after decellularization and scattered throughout the sample but were mainly distributed in the center of the ovaries. Immunohistochemical investigation of the collagen fibers revealed that their contents were reduced (Figure 2C and Appendix A). Interestingly, the Dec tissues contained even higher collagen concentrations than the native tissues (Figure 2D). Alcian blue staining revealed a substantial reduction of acid mucopolysaccharides (Figure 2G). Periodic acid–Schiff staining also suggested the retention of glycosaminoglycan (GAG) (Figure 2G). The ECM components of other decellularization strategies were also evaluated (Appendix A and Appendix A). Therefore, the decellularization effectively eliminated the ovarian cellular components and retained the ECM.

To analyze the variation of collagen fibers, which comprise the main components and structure among the ECM proteins, the CT-FIRE program was used to automatically recognize and quantify the collagen fiber parameters under polarized light images (Figure 2E and Appendix A). Regardless of their respective types, the total collagen fibers were generally arranged parallel to the outer surface and were enriched in the cortex, while the fibers surrounding the GCs and follicles were weaker. The collagen fiber length shortened slightly after decellularization, and we believed that it was acceptable considering the practical sense (Figure 2F). The two groups had comparable collagen fiber width and straightness. Collectively, the results indicated that decellularization partly preserved ovarian ECM protein components and collagen fiber structure.

The good preservation of structural and soluble ECM proteins is important for cellular attachment and signaling transduction. We performed immunohistochemistry for ECM proteins to investigate the ECM more accurately. COL1A1 was located in the tunica albuginea and previous regions of theca cells and medullary stroma. COL2A1, COL3A1, and FN were observed throughout the cortex before and after decellularization. LAMA1 and TGFB1 were mainly localized in the blood vessels. AMH existed in the basal lamina of the previous ovarian surface epithelium and the previous surrounding region of follicles. BMP15 was abundant in regions that were previously occupied by GCs and stromal cells. Semi-quantification of the protein levels revealed that all proteins except TGF-β were decreased to various degrees after decellularization (Figure 2H). Altogether, our results suggested that most of the ECM proteins were preserved after decellularization and their structure and distribution were not disrupted.

### 3.3. Measurement of Cellular Toxicity of Dec Scaffolds via GC and Follicle Culture

We cultured murine GCs on the scaffolds to determine the scaffold cytocompatibility. We isolated and purified primary murine GCs and identified GCs with the markers CYP19A1 and FSHR by immunofluorescence staining (Appendix A). Then, we seeded the GCs onto the sterile scaffolds. After 7 days, adherent GCs were observed on the scaffold surfaces and some GCs had migrated into the inner sites (Appendix A). In some regions, the GCs gathered in clusters (Appendix A). After 15 days, the stereo structure of the GCs was preserved and they continued to stain positively for CYP19A1 (Figure 3A,B), which was indicative of hormone synthesis. Furthermore, most GCs underwent proliferation (Figure 3C) and few apoptotic cells were observed (Figure 3D). Herein, the scaffolds were demonstrated to support GC survival and growth for at least 15 days.

Next, we attempted to culture more complicated components, especially follicles with a diameter > 100 μm, on the scaffolds. The follicles retained their round shape and compact density (Figure 3E) on day 1, with conditions similar to that of native follicles in vivo. However, most of the follicles collapsed after 7 days and lost their spatial structure (Figure 3F). The GCs were also loosely connected with each other. Collectively, the scaffolds could support GC survival for at least 15 days but might not be able to sustain follicle growth for 7 days.

### 3.4. Transplantation and Functional Evaluation of Dec Scaffold-Based Artificial Ovaries

Since adequate efficiency and good biocompatibility were demonstrated, we further evaluated the function of the artificial ovaries in vivo. We constructed artificial ovaries by adding ovarian cells and follicles of all stages onto the Dec scaffolds and transplanted them into submucosal sites in mice in the Sham, OVX, blank, and graft groups (Appendix A). The mouse vaginal orifice and estrus cycle were recorded daily for the following 4 weeks (Figure 4A,B). Sham group mice maintained their regular estrus cycle, and neither the OVX nor blank group mice had restored estrus cycles (Figure 4C). However, the estrus cycles of the graft group mice did not recover as expected, and the vaginal openings of the graft mice also remained closed. Taken together, these findings suggested that the grafts lost their function in the mice in vivo.

To explore the reasons for the poor performance of the artificial ovaries, we retrieved the transplanted grafts on days 3 and 7 (Figure 5A). There was severe inflammatory cell infiltration at the transplantation sites in most mice (Figure 5B). On day 3, the immune cells clustered around the grafts and even grew into them (Figure 5B). Additionally, the inflammatory cells surrounded and destroyed some follicles (Figure 5C). On day 7, an inflammatory encapsulation formed and only some scaffold remnants could be observed (Figure 5D,E). At week 4, half of the grafts had disappeared and we could not retrieve any scaffolds. Immunofluorescence of the day-7 grafts revealed that complement component C3 was localized at the graft periphery, which might have played a central role in the complement system activation (Figure 5F). Monocytes or macrophages expressing CD14 were also detected in the marginal areas. However, strong fluorescence intensity for CD18 and H2AB1 was observed centrally in the scaffolds, which indicated the presence of immune response and antigen-presenting cells, respectively (Figure 5G). The number of CD3 positive cells were few (data not shown). Therefore, the transplantation of artificial ovaries mainly caused the innate immune response.

To reveal the source of the immune response, we investigated the residual substance of the organelles, as they might cause immune rejection. Through the specific marker of respective organelles, we demonstrated that peroxisome, lysosome, ribosome and cytoskeleton were absent after decellularization (Appendix A). However, some residues of mitochondrion and endosome still remained in Dec tissues. These substances might further cause immune rejection and compromise the function of grafts.

## 4. Discussion

The artificial ovary is an effective approach to overcome the adverse effects resulting from ovarian dysfunction. Among the supporting materials for constructing artificial ovaries, Dec scaffolds have the advantages of having adequate ECM components and suitable feasibility, leading to the artificial organs resembling the natural organs more closely. In the present study, we successfully constructed ovarian-specific Dec scaffolds and confirmed their high compatibility in vitro. We constructed the artificial ovaries using Dec scaffolds and transplanted them into a mouse model. Although the scaffolds did not restore ovarian function, they are a reminder of the important role of the immune response during artificial organ application.

We chose porcine ovaries because the duration of swine follicular development and tissue size are closer to those of healthy women than those of other mammals, such as cattle and sheep. Moreover, swine ovaries are easily accessible, so the decellularization protocol may be applied to human ovaries [17]. Human ovaries are the most valuable samples on which to perform decellularization. Typically, they are collected from transgender people or donated by patients undergoing ovarian tissue transplantation, which makes it difficult to obtain human samples [13,16]. Decellularizing mouse ovaries damaged collagen fibers, and thus we did not adopt this approach [15]. 

The purpose of decellularization is to construct artificial organs for clinical application. The balance between cell clearance and ECM protein retention should be maintained. An optimized decellularization protocol should consider tissue density, thickness, porosity, and function [18]. Accordingly, we decellularized ovaries using combined multiple decellularization strategies (mechanical, chemical and biological reagents). Other research groups have also used combined methods, among which SDS, Triton X-100, and SDC were the most frequently used [15,19]. Other reagents such as sodium lauryl ester sulfate, NaOH, and propanol-2 were also used to decellularize ovaries [12,20,21]. However, comparisons of the efficiency and cytotoxicity are limited.

We intended to characterize several methods that target large samples. The cell debris in “M” might be attributed to low concentration of SDS [22]. Our studies also found that different reagents could selectively damage ECM. “W” might damage acidic glycosaminoglycans due to the existence of surfactants, while “P” destroyed neutral glycosaminoglycans because of sodium hydroxide. But this reduction could also be beneficial. For example, a slight decrease in GAG would reduce the density of ECM, thus allowing cells to colonize, adhere, and penetrate better [23].

The freeze–thaw cycle is always the initial step during decellularization. The cells swell from internal hypotonic fluid at low temperatures [24]. Triton X-100 is a non-ionic surfactant that disrupts lipid–lipid and lipid–protein connections without affecting protein–protein interactions. Similarly, SDS is an ionic surfactant and exerts a more intense disruption effect on lipids, proteins, and nucleic acid [25]. Sistani et al. tested 0.1–1.5% SDS and concluded that 72-h treatment of ovarian tissues with 0.5% SDS was adequate to achieve successful decellularization [26]. The surplus DNA was further digested in RNase/DNase solution. Other groups also frequently used trypsin-EDTA [15,21,27]. In sum, our protocol was sufficient to decellularize ovarian tissues.

ECM proteins are the functional basis of Dec scaffolds. We determined that they were inevitably attenuated after decellularization. However, we detected a significant increase in collagen concentration. This might be due to the higher density of parenchymal cells in the native tissues and after removing the cellular components, the ECM protein concentrations increased instead. Collagen fibers contain alkaline groups and double refractive properties, and the acidic PSR dye can react with collagen chains, which causes the latter to become heterochromatic under polarized light [28]. CT-FIRE is a MATLAB software framework developed by Liu et al. [29]. It can extract individual fibers through fiber tracking algorithms and directly identify the optimal curve edges through curved wave transformations, which renders the collagen fiber analysis more precise and time-consuming than previous manual calculations. Considering its actual environment, we thought the reduction in fiber length made little difference to its structure. Therefore, we believed that decellularization exerted fewer effects on the collagen fiber structure.

The observation that GCs grew on the Dec scaffolds for 15 days suggested the low cytotoxicity of the tissues, i.e., surfactants and enzyme reagents were removed completely after subsequent washing. The longest time for cell growth was 33 days, as reported by Pors et al., and the cells achieved complete recellularization [16]. In the present study, most follicles lost their shape after 7 days. We supposed that the Dec scaffolds were unable to provide sufficient three-dimensional adhesion sites for follicles with a diameter > 100 μm and this resulted in loose and abnormal cell conjunction. Our result was similar to that found for the ovarian ECM-based hydrogel by Chiti et al., who reported that the ECM hydrogel was too soft to maintain the spherical shape of follicles [15]. Alternatively, the collapsed morphology might be due to the inability of follicles to remodel the porcine ECM. As the follicles were enclosed by the basal lamina, the secreted proteases were unable to promote the remodeling of scaffolds, thereby reducing the performance of the follicles. Taken together, Dec scaffolds could support the survival of GCs and follicles in vitro, but the function of follicles might be compromised. The addition of stromal cells or surface coating with active and adhesion components, such as Arg-Gly-Asp peptides and poly (ethylene glycol) might enhance recellularization.

Ovarian somatic cells, including embryonic fibroblasts, mesothelial stem cells, macrophages, theca cells and GCs, have been used to co-culture with follicles [21,30,31,32]. In the in vivo experiments, GCs were in-cooperated into the scaffolds to preferentially improve the quality of early-stage follicles (primordial and primary follicles). Because it is easy to lose the critical physical association between oocytes and GCs, more GCs may ensure adequate contact with early-stage follicles [33]. Moreover, as in vitro culture of the follicles exhibited flatted and scattered morphology, some oocytes escaped from surrounding GCs. The addition of GCs provides these oocytes with essential signals and nutrition, which mimics the culture system of GC-oocyte complexes [34,35]. In order to improve follicle survival, the addition of other somatic cells should be considered in future studies.

Currently, many more ovarian-derived Dec scaffolds have been constructed without robust in vivo evaluation, which is considered to be the most critical assessment of artificial ovaries. In the present study, the graft group mice did not resume estrous cycles nor did vaginal openings form, and this indicated that the artificial ovaries did not execute function, which might result from inflammatory cell infiltration. Immune cells could cause damage to follicles and shorten the existing time of grafts. Our findings agreed with those of Liu et al., who demonstrated the involvement of the macrophage response against Dec scaffolds [36]. By contrast, another group reported that the immune cells resulted from Dec tissues being unable to cause a pathologic response [9]. 

Our results suggest the activation of innate immunity around the implanted sites, which has also been reported in other studies [9,37,38]. When the Dec corneal matrix was implanted subcutaneously, an increase of CD68+ macrophages was observed. The implantation of Dec pericardium also caused the infiltration of macrophages [39]. To avoid immune response in mice, therapeutic inhibition that targets specific molecules of the innate immunity, such as C3 and CD14, can be utilized [40]. 

The immune rejection might be attributed to the following factors: (1) the heterogeneity within tissues. The density and compartment within an organ can influence the efficiency of decellularization. Therefore, the implanted scaffolds might contain some nucleus substances. (2) Residual organelle debris and lipids. Our study demonstrated the residual components of mitochondrion and endosome, and they were reported to cause immune response [41]. (3) Although most ECM proteins are conserved among species, some unique protein episodes within porcine ovaries may cause immune rejection [42]. (4) The alteration of ECM structure. The perturbation of collagen structure (morphology, collagen alignment, and thermal stability) can trigger the innate immune response, but its mechanism remains unclear [43]. In terms of ovarian decellularization, some studies showed no existence of inflammatory cells infiltration, while others reported inflammatory cells infiltration. These inconsistent results suggest the need for immunological evaluation of Dec tissues, especially considering that there were various decellularization methods and a lack of in vivo studies.

This study had some limitations. First, recellularization efficiency should be improved. Both direct deposition and injection caused a massive loss of seeding cells (data not shown). Most cells sedimented directly on the cell plate rather than adhering to the scaffolds. Second, immune rejection has long been the focus of organ transplantation and addressing this issue should be a research focus. Experimental settings using immunodeficient mice, immune regulatory regents or even crosslinking agents are necessary [44]. However, the underlying immune mechanism of artificial ovary transplantation was not fully investigated. Finally, the Dec scaffold was constructed using porcine ovaries, which may not fully reflect the true state of human ovaries.

## 5. Conclusions

We proposed an effective decellularization protocol to prepare ovarian scaffolds, which exhibited the characteristics of few nuclear substances and well-preserved ECM proteins. The scaffolds were able to support cellular attachment and growth. However, the artificial ovaries caused a severe immune response in mice in vivo, which suggested that immune rejection plays an important role when using artificial ovaries. Additional studies are necessary to determine the suitability of Dec scaffolds.

## Figures and Tables

**Figure 1 jfb-13-00165-f001:**
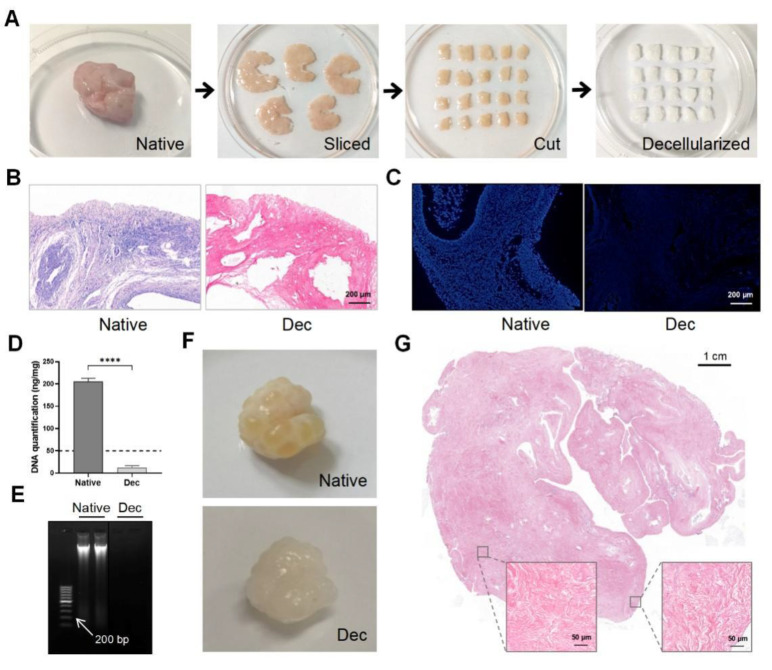
Preparation and evaluation of Dec scaffolds. (**A**) The decellularization process of porcine ovarian cortical strips. Ovaries displayed comparable shape and homogeneity after decellularization, and the color turned from red to white. (**B**) H&E staining showed the presence of both basophilic (cell nuclei) and eosinophilic (cell cytoplasm and ECM) materials in native tissues, while cell nuclei were absent in Dec tissues. (**C**) DAPI staining displayed the presence of cell nuclei in native ovaries, which disappeared after decellularization. (**D**) DNA quantification demonstrated a significant decrease in DNA content of the Dec scaffolds compared to the native tissues. (**E**) The dsDNA fragments of Dec tissues were shorter than 200 bp. (**F**) The appearance of the whole ovarian tissues before and after decellularization. (**G**) H&E staining showed the absence of cell nuclei in the cortical and medullar regions in decellularized tissues. **** *p* < 0.0001. Abbreviations: bp, base pair; DAPI, 4′,6-diamidino-2-phenylindole; Dec, decellularized; ECM, extracellular matrix; H&E, hematoxylin-eosin.

**Figure 2 jfb-13-00165-f002:**
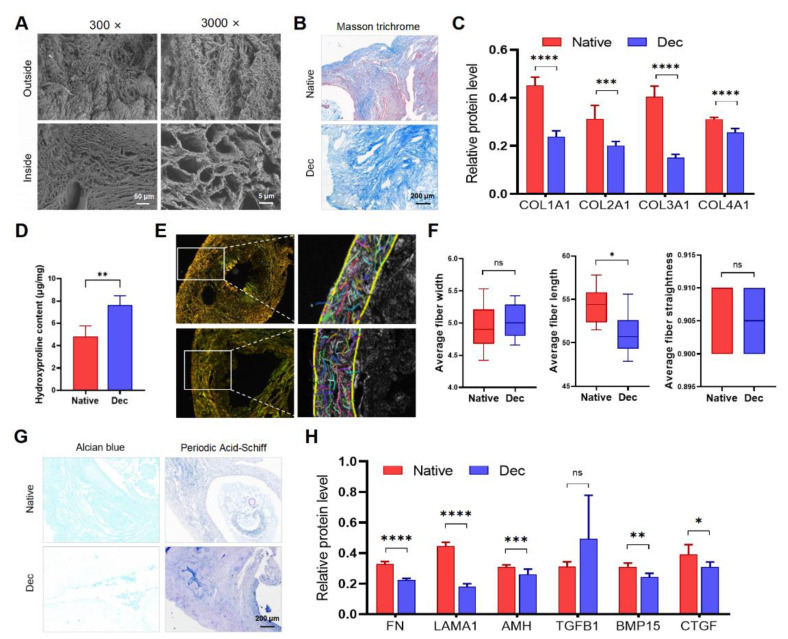
The constitution of decellularized ovarian scaffolds. (**A**) Scanning electron micrographs of native and Dec ovaries. Complex fiber networks with porous structures were visible from both outside and inside of Dec tissues. (**B**) Masson trichrome staining showed the persistence of collagen fibers (blue) and their comparable distribution. The cellular components (red and claret) were removed. (**C**) Semiquantitative analysis of collagens using IHC staining. (**D**) Hydroxyproline assays demonstrated a significant increase in collagen concentration after decellularization. (**E**) Polarized light micrographs of native and Dec ovaries (left). Collagen fibers were indicated by colorful short lines using FIRE-CT program in MATLAB (right). (**F**) The width, length and straightness were compared between native and Dec tissues. (**G**) Alcian blue and PAS staining revealed the loss of GAG and preservation of acid mucopolysaccharide in Dec scaffolds. (**H**) Semiquantitative analysis of selected ECM proteins. * *p* < 0.05, ** *p* < 0.01, *** *p* < 0.001, **** *p* < 0.0001, ns, not significant. Abbreviations: AMH, anti-Müllerian hormone; BMP, bone morphogenetic protein; COL, collagen; CTGF, connective tissue growth factor; FN, fibronectin; GAG, glycosaminoglycan; IHC, immunohistochemistry; LAMA, laminin A; PAS, periodic acid-Schiff; TGFB, transforming growth factor-beta.

**Figure 3 jfb-13-00165-f003:**
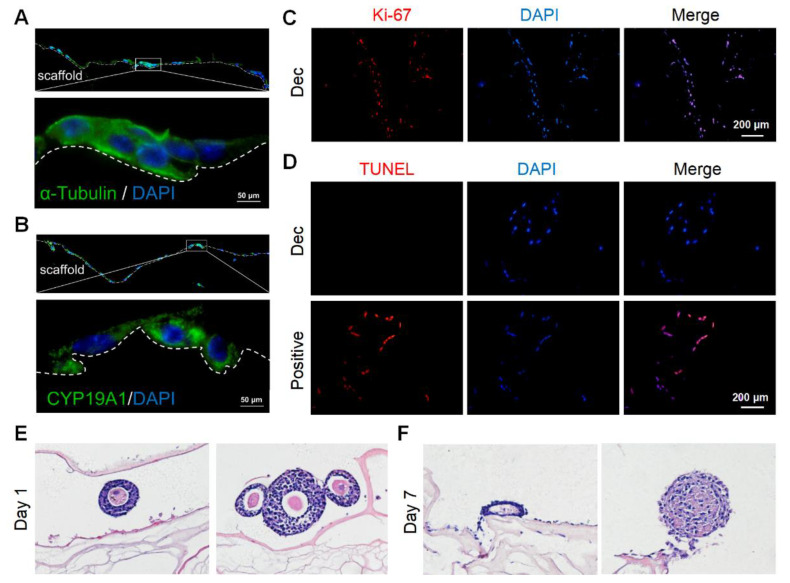
Recellularization of Dec scaffolds with ovarian cells and follicles. (**A**) Cytoskeleton α-tubulin staining demonstrated the stereoscopic structure was well-preserved on day 7. (**B**) CYP19A1 indicated the steroid hormone synthesis function of seeding cells on day 7. (**C**) Ki-67 staining confirmed the presence of proliferating cells on day 15. (**D**) Representative images of the TUNEL assay showed few apoptotic ovarian cells on day 15. H&E staining of follicles cultured on Dec scaffolds on day 1 (**E**) and day 7 (**F**). Abbreviation: CYP19A1, cytochrome P450 family 19 subfamily A member 1; TUNEL, terminal deoxynucleotidyl transferase-mediated dUTP nick end-labeling.

**Figure 4 jfb-13-00165-f004:**
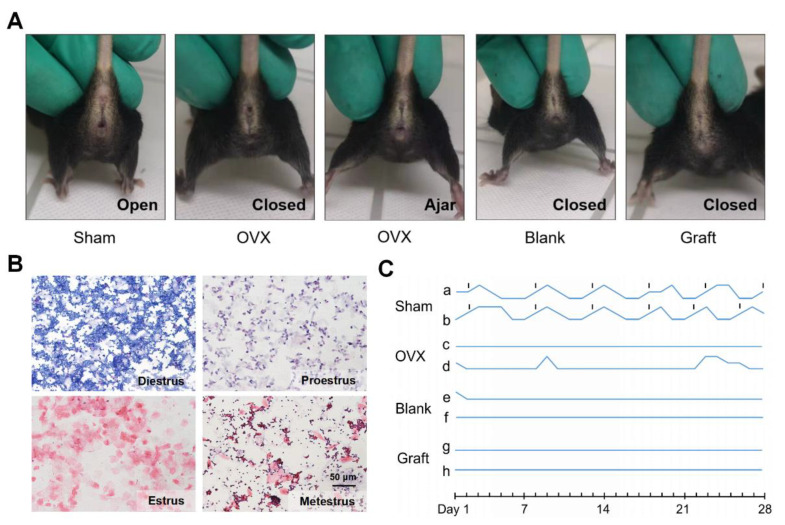
In vivo function of the transplanted artificial ovaries. (**A**) The condition of vaginal opening in different groups. (**B**) Representative images of vaginal smear after transplantation indicated the estrus cycles of mice. (**C**) The dynamic variation of estrus cycles of mice after 4 weeks.

**Figure 5 jfb-13-00165-f005:**
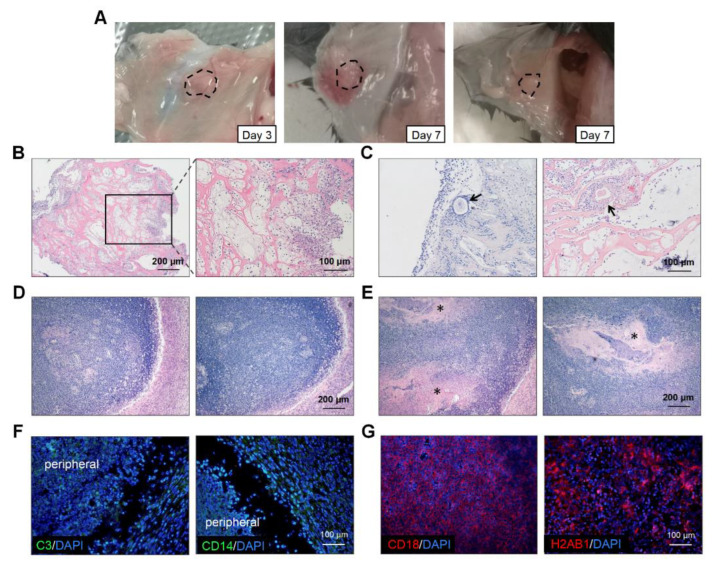
Immune response assessments of transplanted grafts. (**A**) The macroscopic observation of transplanted scaffolds on day 3 and day 7. (**B**) H&E staining revealed that inflammatory cells were scattered throughout the grafts and some invaded into the scaffolds on day 3. (**C**) Individual follicles (arrowhead) could be seen among these inflammatory cells. The total disappearance (**D**) and partial remnants (**E**), indicated by asterisks, of scaffolds on day 7. (**F**) Fluorescence staining for C3 and CD14 were observed in the cells located peripherally on the scaffolds. (**G**) Cells expressing CD18 and H2AB1 cells could be seen in the central part. Abbreviations: C3, complement C3; CD, cluster of differentiation.

## Data Availability

Not applicable.

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
