# Peer review of "Construction of Artificial Ovaries with Decellularized Porcine Scaffold and Its Elicited Immune Response after Xenotransplantation in Mice"

_jfb, 2022, doi:10.3390/jfb13040165_

Round 1

Reviewer 1 Report

The manuscript by Wu and co-authors describes the attempt to create models of artificial ovaries using decellularized porcine ECM scaffolds in mice.

The work is of potentially high significance, and the authors have conducted a comprehensive body of work to create and characterize decellularized matrix as a model of ovaries.

Critique points:

1. Analyses of residual ECM and ECM-bound proteins shows significant reduction in the amount of residual proteins, while the conclusion (line 270) states that they were preserved, which is a discrepancy.

2. The authors should investigate scientific literature on a potential role of rejection of pig-derived ECM by mice as an explanation for the observed poor performance of their scaffolds.

3. The authors need to investigate whether collapse of the follicles was due to their inability to digest the porcine ECM.

4. Scheme 3A: what are the individual lanes on the gel?

5. Language and style need to be extensively edited.

Reviewer 2 Report

The authors presented the construction method of artificial ovaries using decellularized porcine scaffolds in the paper. They constructed decellularized scaffolds with porcine ovarian tissues and evaluated the structure and morphology of the scaffolds by electron microscopy and immunohistochemistry. Furthermore, they transplanted artificial ovaries with the scaffolds into ovariectomized mice to examine the function of the artificial ovaries in vivo. They found that granulosa cells and follicles attached to the scaffolds were alive on the day 15 and 7, respectively. Overall, the manuscript appears to be well written. However, I have a few major concerns about this article. 

1. The usefulness of decellularized porcine scaffolds which is presented in this paper is not clearly demonstrated. The authors should indicate the usefulness of using porcine tissues or the problems with existing scaffolds in the introduction section.

2. In addition to the above, a comparative study should be added to determine which aspects of the scaffold in this paper are superior to the existing ones, and which aspects are problematic.

3. Because this paper has some typographical and grammatical errors, it needs to be proofread again.

Reviewer 3 Report

In the present study, the AA propose a new protocol of decellularization (DEC) of porcine ovarian biopsy/whole ovaries. After investigating through morphological and molecular processes the efficient removal of cell content from the organ, they aim at using the DEC tissue for reconstituting artificial ovaries with mouse derived granulosa cells (GCs) and follicles. The latest part, in part successful in the in vitro asset, failed to support follicle growth and to recreate a normal environment when xenotransplanted in ovariectomized mice.

Even if the article falls in the hot topic categories of the reproductive field research, it lacks novelty and scientifical resonance in the current form, mainly due to some pivotal mistake in the planning of the experiments, and for this reason cannot be accepted for publication in the current form, in my opinion. Please, see the main points below:

1.     English editing and scientific language: The manuscript needs to undergo an extensive English proofreading, due to the presence of several typos, incomplete sentences (lines 112-113 “morphology and structure of were acquired…”) and unclear phrasing. Also, the AA need to improve the scientifical writing, as numerous times during the manuscript they used an informal way of writing (e.g., lines 88 “stick by syringes” “butterflied”, 95 “they were shook”, 97 “agitated”, 151 “200 ul”, 177 “2.0 ´ 106”, and so on). This lack of attention in the writing is highly disturbing during the reading and leads to misinterpretation or complete misunderstanding of sentences.

2.     Aim, part 2: One of the most concerning issues for me regards the experimental setup of the second part of the manuscript. The AA aim to create an artificial ovary by using the porcine DEC-ovaries as scaffold for the seeding of murine GCs and follicles. By doing so, the AA are adding the double of GCs component to the scaffold (since follicles are composed mainly by GCs, oocytes and theca cells, depending on the stage), without providing to the follicles the presence of supporting stroma or androgen-producing cells. During folliculogenesis, the synthesis and metabolism of androgen is essential for follicle growth and development. Without the presence of this hormone, the folliculogenesis stops at quite early stages (around secondary stage follicle), given the lack of substrate for the GCs-driven production of 17-b-estradiol.

The decision to seed extrafollicular GCs to the scaffolds, is not only scientifically wrong in the perspective of ovarian biology, but also does not improve the chances of a positive result in terms of follicles support (both spatial- and viability-wise). Thus, all the second part of the paper (excluding the GCs for the evaluation of the cytotoxicity) is lacking a scientifical background and therefore outcome.

3.     Growth factor and hormone detection as validation of the decellularization protocol: During the validation of the decellularization, the AA investigate the presence of hormones or growth factors (i.e., AMH, TGFB1, BMP15) in DEC ovaries (lines 264-269). Since the decellularization process is extensively long and undergoes several changes in temperatures and chemical incubations, it is quite challenging the half-life survival of those compounds (mostly considering that there is no cellular compartment able to produce them anymore).

Moreover, how can the AA talk about “surface epithelium”, “surrounding part of the follicles”, “GCs compartment and stromal cells” if those cells are not anymore present in the DEC tissue? And if they mention those localizations, it could be needed a representative picture of each GF and hormone investigated.

4.     Follicle culture method and staging: I have to raise several doubts about this part of the manuscript for the way that the methods are presented and often not explained, and for a further experimental setup mistake that expectedly led to no final result.

Firstly, the AA do never explain the methodology through which they isolate follicles from mice ovaries, neither how they evaluate the maturation stage of the follicle, but they just mention the size (ca. 100 µm). The different stages of maturation are extremely important when coming to the presence or not of theca cells and their activity in producing androgens that, as mentioned already in point number 2, will play a pivotal role in the development and regulation of the same follicle and of the surrounding follicles.

Moreover, the stage of maturation is indispensable for setting up the correct follicle culture media additions, as already explained and documented in several papers regarding in vitro folliculogenesis in mice models (e.g., https://doi.org/10.1262/jrd.2021-091, https://doi.org/10.1371/journal.pone.0099423, https://doi.org/10.1093/humrep/det338). The lack of this knowledge leads to the use of inappropriate culture media during the follicle culture (i.e., McCoy’s 5A + 10% FBS, Androgen, 1% antibiotics). How much of the androgen has been added in culture? And in case the follicles are already producing the hormone autonomously, are the AA sure that the follicles are not already producing the correct amount needed, and by exogenous addition they are creating an hyperandrogenic environment? Why are they not adding other hormones (FSH, E2)?

Unfortunately, those culture conditions are insufficient, and the basis for follicle survival/growth are not met.

5.     Immune response after xenotransplantation: The AA state in the conclusions that “the artificial ovary causes severe immune response in mice in vivo” and that “it suggested the important role of immune rejection when using artificial ovaries”. Since the artificial ovaries were actually produced from porcine ovaries, later on transplanted in ovariectomized mice, the possibility of immune response should have been a starting point, and not the conclusion. Being two very different and taxonomically different species, the AA should have worked on another way of avoiding immune response in mice who underwent xenotransplantation (as the use of immunodeficient mice, or the utilization of another species for the decellularization, or another recipient).

I totally understand and agree on the choice of using porcine ovaries, since, as they state during the discussion, these are the closest morphological- and functional-wise to the humans. But the not well thought application of the very well obtained decellularized ovaries did not achieve the hoped result.

I would like to mention my sincere agreement on the new protocol presented in the manuscript, since this part produced very high-quality results, even if poorly validated.

Based on the previous comments, I cannot recommend this paper for publication.

Round 2

Reviewer 1 Report

My comments were addressed.

Reviewer 3 Report

I would like to thank the authors for the great work made in experimental and writing matters. The current version of the manuscript has been definitely improved and, even if some question couldn’t be experimentally answered, I appreciated the time and effort that they put in explaining pros and cons of their work.

Nevertheless, some points should be considered before publication:

Major:

1.     The title needs to be more comprehensive and accurate of what are the outcomes achieved/not achieved through the construction of these artificial ovaries after xenotransplantation in ovariectomized mice. As it is, it does not give any useful information. Please, amend properly.

2.     Answer to point 2 of the first round of revision: I understand the explanation of the authors, but I still believe this is a noticeable gap in your work. I would highly recommend adding a paragraph in the discussion part in which the author explain the choice of using GCs instead of stromal cells and discuss in a deeper manner the reason why follicles were collapsing. Like this, any criticism to this manuscript could be avoided and the reader would be conscious of the author’s choice in the experimental setup.

Minor:

1.     Line 89: the term “syringing” cannot be used as the authors mean. Please, amend properly.

2.     Line 124: “a microscope” is insufficient. Amend as follows: “an ultra-high resolution scanning electron microscope”.

3.     Line 131: add missing information on Olympus light microscope used (i.e., series and model, as the authors specify in line 219 of the manuscript).

4.     Line 297: replace “stromal” with “stroma”.

5.     Lines 372-373: remove dECM as abbreviation, since the term is not used in the figure nor in the legend.

6.     Line 377, 383: replace “microphotograph” with “micrograph”.

7.     Line 390: “LAMA: Laminin A”.

8.     Line 398: Please, add the day of culture in which you find TUNEL positive cells, as mentioned for all the other pictures of the panel.

9.     Line 469: The table adds a great value to prove the efficiency of the protocol that the authors apply. Please, add the years next to each reference, and it would be helpful to add an extra column in which is reported the species used for decellularization.

10.  Line 570: replace “conservative” with “conserved”.

11.  Line 574: add the period after reference [38].
